# Remote and Proximal Sensing Techniques for Site-Specific Irrigation Management in the Olive Orchard

Giovanni Caruso [1], Giacomo Palai [1,*], Riccardo Gucci [1] and Simone Priori [2]

1   Department of Agriculture, Food and Environment, University of Pisa, Via del Borghetto, 80, 56124 Pisa, Italy; giovanni.caruso@unipi.it (G.C.); riccardo.gucci@unipi.it (R.G.)
2   Department of Agriculture and Forest Sciences, University of Tuscia, Via S.C. de Lellis, 01100 Viterbo, Italy; simone.priori@unitus.it
*   Correspondence: giacomo.palai@phd.unipi.it

**Abstract:** The aim of this study was to evaluate the potential use of remote and proximal sensing techniques to identify homogeneous zones in a high density irrigated olive (*Olea europaea* L.) orchard subjected to three irrigation regimes (full irrigation, deficit irrigation and rainfed conditions). An unmanned aerial vehicle equipped with a multispectral camera was used to measure the canopy NDVI and two different proximal soil sensors to map soil spatial variability at high resolution. We identified two clusters of trees showing differences in fruit yield (17.259 and 14.003 kg per tree in Cluster 1 and 2, respectively) and annual TCSA increment (0.26 and 0.24 dm$^2$, respectively). The higher tree productivity measured in Cluster 1 also resulted in a higher water use efficiency for fruit (WUE$_f$ of 0.90 g dry weight L$^{-1}$ H$_2$O) and oil (WUE$_o$ of 0.32 g oil L$^{-1}$ H$_2$O) compared to Cluster 2 (0.67 and 0.27 for WUE$_f$ and WUE$_o$, respectively). Remote and proximal sensing technologies allowed to determine that: (i) the effect of different irrigation regimes on tree performance and WUE depended on the location within the orchard; (ii) tree vigour played a major role in determining the final fruit yield under optimal soil water availability, whereas soil features prevailed under rainfed conditions.

**Keywords:** NDVI; *Olea europaea* L.; precision orchard management; soil apparent electrical conductivity; tree water status; unmanned aerial vehicle; water use efficiency

## 1. Introduction

Water availability is the main limiting factor for the growth and yield of crops in the Mediterranean region, which is expected to undergo dramatic changes in temperature and precipitation due to climate change [1,2]. Climate change will also likely increase the frequency of heat waves and prolonged periods of drought [3,4], inducing uncertainties about the potential productivity of crops. Irrigation can help to mitigate the impact of climate change on perennial crops, including olive trees. Although olive trees are drought tolerant, many studies showed the beneficial effects of irrigation on vegetative growth, yield components, and oil quality [5–10].

An important aspect in irrigation management is to understand the multiple interactions between water availability and soil properties or canopy vigour, which may potentially cause variations in yield and vegetative growth. In this respect, site-specific management of inputs, such as fertilizers and irrigation water, requires the understanding of the spatial distribution of soil characteristics, tree vigour, and productivity. Homogenous zones within the orchard for tree vigour and soil properties can be identified by soil sampling protocols and by monitoring parameters of vegetative growth (e.g., canopy volume increment, trunk growth rate), albeit at a high cost of money and labor [11,12]. Recently, Moral et al. [13] developed a predictive model based on soil features to identify homogeneous zones within olive orchards, but the precision and the accuracy of these maps were strictly related to the number of soil samples analysed. The use of remote and proximal sensing technologies

for estimating field variability is becoming more and more common in precision agriculture due to their relatively lower cost and the non-invasive approach with respect to conventional methods [14–16].

Among proximal soil sensing (PSS) technologies, electromagnetic induction (EMI) has been the most commonly used for the last 20 years [17,18]. Several authors used EMI sensors to map soil features and homogeneous zones in vineyards and orchards [19–21]. These sensors measure the apparent electrical conductivity (ECa), which is closely related to soil texture, coarse fragments, salinity, and moisture [17]. Mobile gamma-ray spectroradiometry is an additional promising PSS technique for precision agriculture [22–24]. Gamma-ray spectroradiometry measures natural $\gamma$-emissions from the main radionuclides (40 K, 238 U, and 232 Th) and the total emissions from all elements in the soil depth of the first 0.3–0.4 m. The concentrations of these $\gamma$-rays emitters are strongly related to the mineral composition of the parent material, but also to clay percentage and superficial stoniness [22]. In the absence of soil salinity, ECa responds primarily to the amount of clay, moisture, and bulk density, while gamma-ray spectroradiometry is more influenced by the variation in parent material mineralogy [14].

Homogeneous zones for precision orchard management can also be identified based on geometrical and spectral canopy information derived from aerial images. Remote sensing techniques have recently been shown to be effective in estimating tree vigour and canopy geometry as an alternative to the on-ground field measurements in orchards and vineyards [25–29]. In previous studies aerial images and spectral indices were used to determine the canopy projection in the olive orchard and to derive relationships with soil and tree structures to be applied in site-specific management [30,31]. Recent studies also focused on predicting yield based on geometrical and spectral canopy characteristics [32,33].

Despite the advantages provided by the above-mentioned technologies for soil and crop sensing, different authors suggest a combined use of remote and proximal sensing data as a much more reliable approach to identify homogeneous zones in orchards and vineyards [34–36]. Cluster analysis of maps obtained both by soil survey, such as proximal soil sensing, and by plant water status, such as NDVI, allowed to identify functional homogeneous zones (fHZs), corresponding to areas where soil and plant performance showed limited variability [37,38]. Soil and crop indices have been mainly tested in vineyards in experiments focused on zoning and precision irrigation management [19,38–42] while few studies have been conducted in orchards [43–45]. To our knowledge, there is only one study in which the use of proximal and remote sensing techniques was used to separate management zones within an irrigated olive orchard [46]. Moreover, there is little information on how the interaction between irrigation, tree vigour, and soil properties affects yield and tree growth in olive. Since water status is one of the most important factors determining plant performance, we manipulated soil water availability to better define how remote and proximal sensing can help to manage irrigation.

The objectives of this study were: (i) to evaluate the effect of different irrigation regimes on tree productivity and growth in different management zones identified using remote and proximal sensing techniques; (ii) to discriminate the effect of tree vigour (NDVI) and soil features derived by proximal sensing on fruit yield and vegetative growth under different conditions of soil water availability.

## 2. Materials and Methods

### 2.1. Plant Material and Site

Experiments were carried out in a high density (513 trees per hectare), 17-year old, irrigated olive orchard at the experimental farm of the University of Pisa, in Venturina (Italy). The orchard included six cultivars, but the work was carried out only on cv. Frantoio, which was the most abundant (61% of the total). A complete description of the plant material and orchard management has been previously published [5,7].

Climatic conditions during the study period were monitored using an iMETOS IMT 300 weather station (Pessl Instruments GmbH, Weiz, Austria). The annual reference evapo-

transpiration (ET0), calculated according to the Penman-Monteith equation, was 952 mm. Annual and summer (21 June–22 September) precipitations were 809 and 39 mm, respectively. Cultural practices and monitoring of phenological parameters, including the estimation of the full bloom date (DOY 148), were performed as previously reported [5,7].

### 2.2. Irrigation Management and Measurements

The trees were irrigated using a subsurface drip irrigation system (2.3 L h$^{-1}$ pressure-compensated drippers spaced at 0.6 m). Irrigation volumes were calculated on the basis of the effective evapotranspiration (ET$_c$), using a crop coefficient (Kc) of 0.55, 0.60 and 0.65 in July, August and September, respectively. The irrigation period lasted from DOY 188 (7 July) to DOY 274 (1 October), during which fully-irrigated (FI) and deficit-irrigated (DI) trees received a total volume of 2160 and 970 m$^3$ ha$^{-1}$, respectively, whereas the rainfed (RF) ones received only precipitation. Fertigation was used to supply mineral nutrients in spring before irrigation treatments started.

Tree water status was monitored at three dates (DOY 187, DOY 218 and DOY 245) by measuring the stem water potential (SWP) on six trees per irrigation treatment (three trees per each cluster-irrigation combination, as explained in Section 2.5). SWP was measured after blocking transpiration of leaves inserted near the main scaffolds of the tree [47].

Vegetative growth was evaluated as the annual increment in the trunk cross-sectional area (TCSA). The TCSA was calculated from the circumference of the trunk at 0.40 m from the ground, measured on DOY 11 and 356. Immediately before harvest, 100 fruits were randomly sampled from around the canopy of three trees per each cluster-irrigation regime combination to measure mesocarp and endocarp fresh and dry weight. At harvest (DOY 285), each tree was harvested by hand and the final fruit yield was weighed. The oil content of the fruit mesocarp of 20 fruits per tree previously sampled for fresh weight determinations was measured at harvest by nuclear magnetic resonance Oxford MQC-23 analyzer (Oxford Analytical Instruments Ltd., Oxford, UK). The mesocarp was cut in small pieces (2–5 mm) and oven-dried at 70 °C for 48 h. The oil yield of individual trees was calculated as previously reported [48]. Water use efficiency for fruit (WUE$_f$) and oil (WUE$_o$) was calculated as the ratio between fruit yield (dry weight) or oil yield, respectively, and annual ET$_c$ within each irrigation treatment.

### 2.3. Multispectral Imagery Acquisition and Analysis

Images were acquired one day before the beginning of the irrigation differentiation (DOY 187) using a multispectral camera (Tetracam ADC-lite, Tetracam, Inc., Gainesville, FL, USA), mounted on a S1000 UAV octocopter (DJI, Shenzhen, China) able to fly autonomously over a predetermined waypoint. NIR-RG images were recorded in the visible red (R), green (G), and near-infrared (NIR) domain with nominal wavelengths of 520–600, 630–690, and 760–900 nm, respectively.

Images were acquired at noon under clear sky conditions, the flight altitude was 70 m above ground level (AGL), as reported by Caruso et al. [25]. Before the UAV flight, a set of eight ground control points (GCPs) were placed in the orchard and georeferenced using a Leica GS09 real-time kinematic GPS (Leica Geosystems A.G., Heerbrugg, Switzerland). Multispectral images were first mosaicked using Autopano Giga 3.5 Software (Kolor SARL, Challes-les-Eaux, France), then georeferenced and orthorectified using the ground reference points (ArcGIS software, ESRI, Redlands, CA, USA) ad reported by Caruso et al. [25]. The NDVI index [49] was calculated using the map algebra technique implemented in ArcGIS software (ESRI, Redlands, CA, USA) using the following equation:

$$NDVI = (NIR - RED)/(NIR + RED)$$

where NIR and RED are the reflectance values in the near-infrared (630–690 nm) and red (760–900 nm) bands, respectively.



### 2.4. Proximal Soil Sensing and Soil Analysis

Two different proximal soil sensors (PSS) were used to map soil spatial variability at high resolution. The proximal sensors used were: (i) EM38-Mk2 electromagnetic induction sensor (Geonics Ltd., Mississauga, ON, Canada); (ii) "The Mole", gamma-ray spectro-radiometer (Soil Company, Groningen, The Netherlands). The EM38-Mk2 measures the apparent electrical conductivity (ECa) of the soil at two depth ranges of about 0–0.75 ($ECa_1$) and 0–1.50 m ($ECa_2$) [50]. Soil physical properties, such as texture, stoniness, bulk density [51], soil moisture and water availability [52,53], soil depth [54], as well as organic matter [55] all affect ECa. "The Mole" spectroradiometer continuously measures the natural gamma-ray emission coming from the first 0.3–0.4 m of the soil and rocks, through a Cesium Iodide scintillator crystal [56]. The gamma-ray spectra were analyzed by a Full Spectrum Analysis (FSA), using "The Gamman" software (Medusa Systems, The Netherlands) [22]. This method identifies and deletes data outliers, as well as processes gamma-ray spectrum for calculation of individual nuclide concentrations (40 K, 238 U, 232 Th) and total counts (TC_gamma) in $Bq \cdot kg^{-1}$. TC_gamma depicts the total count measured by the CsI scintillator for the whole range of energy bands. Proximal gamma-radiometrics has been used to survey topsoil features, such as texture [57], gravel content [22], potassium [58]), and organic carbon [59]. Both the gamma-ray spectroradiometer and EM38-Mk2 were manually driven (without using a tractor or ATV), since the size of the experimental field was relatively small. The sensors were supplied with GPS and rugged PC for data-logging. Soil sensing was performed continuously along each tree row, with the sensors at 0.2 m height above the soil surface, to minimize the signal attenuation. The data were interpolated across the whole area using ordinary kriging (OK), which is widely used or the interpolation of soil proximal sensing data [14,17,19]. The OK parameters, namely lag size, number of lags, and maximum range were selected in order to minimize the estimation error.

Six soil profiles were dug to account for the variability across the field, in terms of proximal soil sensing maps and irrigation management. The soil profiles were described following the international guidelines for soil description [60] and classified according to the Word Reference Base for Soil Resources [61]. Soil samples were collected from several genetic horizons of each soil profile. The samples were air-dried, sieved to 2.0 mm, and analysed for physical and chemical properties using standard laboratory methods. In particular, soil texture was determined by the Micromeritics Sedigraph analyser [62], total organic C (TOC) and total N (TN) were measured by dry combustion with a ThermoFlash 2000 CN soil analyser, after removal of carbonates by HCl 10%. The total equivalent $CaCO_3$ content was calculated from the difference between the total C measured by dry combustion in the untreated soil (mineral C + organic C) and in the HCl-treated soil (organic C). Soil pH was measured potentiometrically in a 1:2.5 soil–water suspension, whereas electrical conductivity was measured in a 1:2 soil–water filtered extract after 2 h shaking and overnight standing. Available water capacity (AWC, in $cm\ cm^{-1}$) was calculated for all the soil horizons using the pedotransfer function of Saxton and Rawls [63] implemented in the software SPAW (U.S. Department of Agriculture, Washington D.C., MD, USA).

### 2.5. Experimental Design

The orchard consisted of 12 plots, each containing 12 trees arranged in three rows of four trees, with the only exception of one plot of two rows [5,6]. Three plots were fully irrigated (FI), six plots were deficit irrigated (DI), whereas the remaining three did not receive any irrigation (RF, rainfed) (Figure 1).

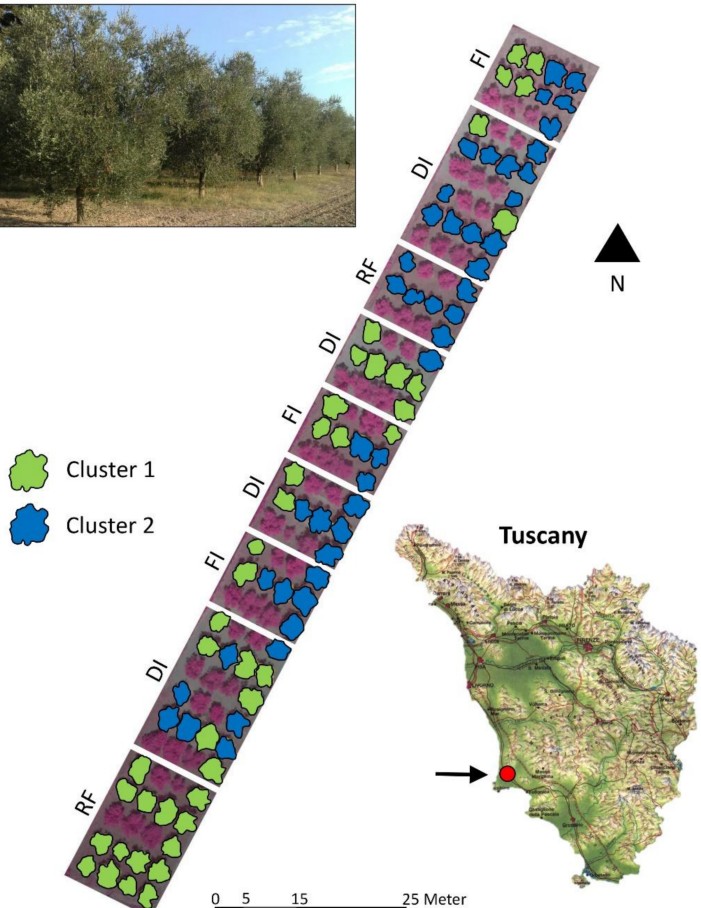

**Figure 1.** False colour orthophoto of the olive orchard located at the experimental farm of the Department of Agriculture Food and Environment of the University of Pisa (43°10′ N; 10°36′ E). The canopies of the olive trees cv. Frantoio are in green (Cluster 1) or light blue (Cluster 2). The trees of other cultivars were not used in the experiment.

The orchard was subdivided into two homogeneous zones (C1 and C2) by k-means clustering, using the Hill-Climbing algorithm [64] of the SAGA-GIS software. The input data to calculate k-means clustering were: $ECa_1$, $ECa_2$, TC_gamma, and NDVI. Such homogeneous zones simulated two prescription areas for site-specific management (in particular precision irrigation) of the olive orchard (Figure 2). The selection of only two clusters was driven by the small surface of the experimental field and by the relatively homogeneous soil.

### 2.6. Statistical Analysis

The SWP was measured in six trees per irrigation regime (three trees per cluster-irrigation combination) and the means were separated by the least significant differences (LSD $p > 0.05$) after analysis of variance (ANOVA). An exploratory correlation analysis was performed between NDVI, soil proximal sensing ($ECa_1$, $ECa_2$, TC_gamma), fruit yield, and TCSA increment, to highlight the statistical relationships between the main parameters used for this study. Since most of them showed non-parametric distribution, the Spearman's ranks correlation test was used. In particular, correlation models within the different irrigation regimes (FI, DI and RF) have been tested by linear regressions. Full factorial ANOVA was carried out to test the effects of the irrigation regimes categorial factors (FI, DI, RF) and homogeneous zones (C1 and C2), as well as the combinations of these factors. Post-hoc LSD Fisher test was used to determine the significant differences between the means of the analysis of variance. To determine the influence of PSS data and NDVI on fruit yield and annual TCSA increment predictions, forward-stepwise multiple

regressions were calculated for each irrigation treatment (FI, DI, RF). The independent variables used for the regressions were $ECa_1$, $ECa_2$, TC_gamma, and NDVI. The selection of variables by the forward-stepwise regression was carried out on the basis of F-to enter of 3, to avoid a nonsignificant contribution of some variables in the regression. Statistical analyses were performed with STATISTICA 7.0 (StatSoft Inc., Tulsa, OK, USA).

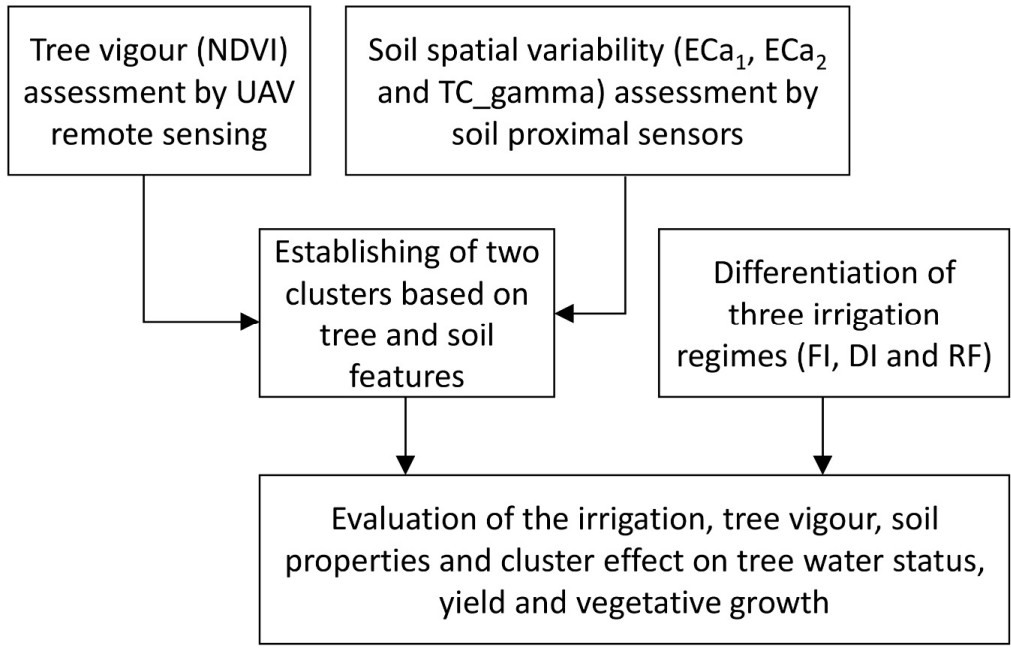

**Figure 2.** Flow chart of the experiment. Legend: NDVI, normalized difference vegetation index; $ECa_1$ and $ECa_2$, apparent electrical conductivity at 0–0.75 and 0.75–1.50 m soil depth; TC_gamma, total count gamma ray; FI, full irrigation; DI, deficit irrigation; RF, rainfed conditions.

## 3. Results

### 3.1. Proximal Soil Sensing and Profile Description

The soil apparent electrical conductivity ranged between 21.6 and 48.0 mS m$^{-1}$ in the 0–0.75 m soil profile ($ECa_1$) and between 33.5 and 55.2 mS m$^{-1}$ considering a deeper soil volume ($ECa_2$, about 0–1.50 m) (Figure 3). The mean values of $ECa_1$ and $ECa_2$ for the entire field were 32.7 and 41.6 mS m$^{-1}$, respectively. Higher values of ECa were measured on the South and East side of the field.

Low variation of gamma total counts (TC_gamma) was measured by gamma-ray spectroscopy. The values ranged between 376 and 442 Bq kg$^{-1}$ and indicated a homogeneous mineralogy of the soil parent material. These TC_gamma values were similar to those measured in other soils developed in mixed fluvial deposits of Central Italy [22]. The same study measured values around 250–300 Bq kg$^{-1}$ in soils developed on calcareous rocks and 550–600 Bq kg$^{-1}$ in soils on feldspathic sandstones [22].

The maps of radionuclides contribution (40 K, 232 Th, 238 U) showed low variability of the values and a patchworked pattern, which made them unsuitable for further statistical analysis. For this reason, the radionuclide maps were not used for the next step of data analysis. Although the variability of the absolute TC_gamma values was low, the map (Figure 3) showed an evident area with lower values (about 380 Bq kg$^{-1}$) in the northern part of the field due to surficial stoniness, characterized by about 10–15% of medium and coarse gravel (1–6 cm size). Surficial stoniness was virtually absent in the rest of the orchard.

Two of the six soil profiles were described in this gravelly area (P1 and P2), and they showed the presence of coarse fragments (from 2 to 12%) up to a depth of 0.90 m, whereas the other profiles showed coarse fragments ≤2% only in the Ap horizons. Table 1 shows the two extremes of the soil spatial variability, characterized by the profiles P1 (gravelly area, lower TC_gamma, lower ECa) and P4 (not gravelly area, higher TC_gamma, higher

ECa). As a result, the field was characterized by deep (1.5 m) and loamy soils, with a slight increase in clay for illuviation in deep horizons (+6%). Thus, it was classified as Calcaric Luvisol (Loamic, Profondic) according to the international WRB classification [61]. The profiles were similar for other characteristics: loamy texture (tending to loamy-clay in deep Bt horizons), low calcium carbonate content (from 2 to 2.5%), sub-alkaline pH (8–8.3) and absence of salinity. The organic carbon (TOC) of the topsoil (Ap1 horizons) varied from 9 to 37 g kg$^{-1}$. The TOC of the subsoil horizons varied between 6.5 (deeper Bt horizons) and approximately 10 g kg$^{-1}$, and the differences between profiles were minimal. Total nitrogen showed the same pattern of TOC, the C/N ratio was relatively stable (approximately 10). Although the soil texture and chemical parameters of the described profiles were quite homogeneous, the different content of the coarse fragments influenced the modelling of the soil available water capacity (AWC).

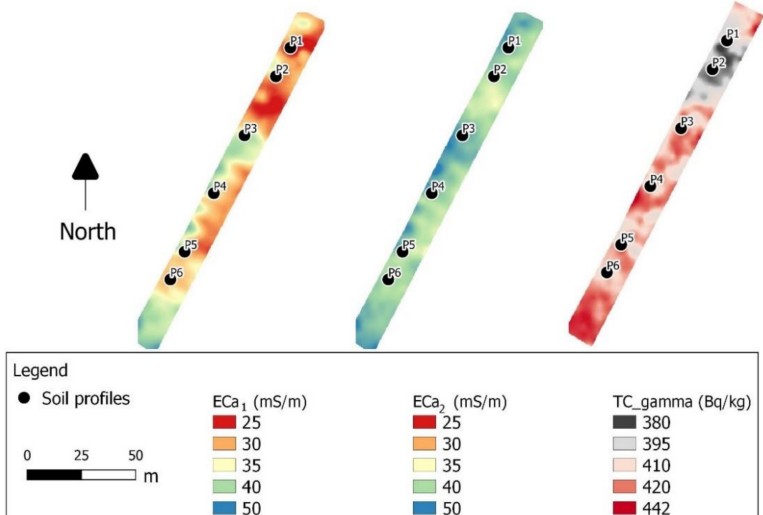

**Figure 3.** The maps obtained by the soil proximal sensing, and the location of the soil profiles (P1–P6, black dots). The left and central maps showed the apparent electrical conductivity (ECa) at two reference depths, about 0–0.75 (ECa$_1$) and 0–1.50 m (ECa$_2$). The map on the right showed the gamma-ray total counts (TC_gamma) of the topsoil, about 0–0.30 m. The lower values of TC_gamma in the northern part of the field correspond to the area with higher surficial stoniness.

According to the pedotransfer function of Saxton and Rawls [63], coarse fragments have a negative coefficient, and therefore tend to decrease AWC. For this reason, soil profiles with 10–15% of gravels showed lower AWC (115 mm m$^{-1}$, for both the profiles) than the other profiles with gravels ≤2% (AWC from 130 to 141 mm m$^{-1}$). The soil profiles also differed in redoximorphic features, which indicated the depth of temporary waterlogging due to the slow deep drainage (stagnic properties). In particular, the soils of the northern part of the field, characterized by slightly lower ECa and TC_gamma (P1 and P2) showed stagnic properties starting at 1.15 or 1.30 m deep. In contrast, the soils characterized by higher ECa and lower TC, in particular in the southern side of the field, showed stagnic properties starting around 0.90–1.00 m deep. Although the difference of 0.15–0.40 m regarding stagnic properties depth is apparently similar, it could influence the water availability for plants. Water saturation at a depth of 0.9–1.0 m for a certain period of the year does not negatively affect root oxygenation but can provide water for plants for a slightly longer time than soil with deeper stagnic properties.

**Table 1.** Description of the benchmark profiles which defined the two areas of the field, gravelly (P1) and not gravelly (P4). Legend: Ap1, surface horizon subjected to ploughing; Ap2, sub-surface horizon interested by deep ploughing; Bt, B horizon with clay illuviation process; Btg, B horizon with clay illuviation process and redoximorphic mottles due to temporary waterlogging (stagnic properties); CF, Coarse fragments; SOC, Soil Organic Carbon; TN, Total nitrogen; AWC, Available Water Capacity.

| Cluster 1 | Soil Horizons | | | |
|---|---|---|---|---|
| (Profile 1) | Ap1 | Ap2 | Bt | Btg |
| Depth (m) | 0.8 | 0.40 | 1.00 | 1.50 |
| Clay (dag kg$^{-1}$) | 13.9 | 22.5 | 17.7 | 22.1 |
| Silt (dag kg$^{-1}$) | 45.5 | 33.2 | 44.4 | 39.3 |
| Sand (dag kg$^{-1}$) | 40.6 | 44.3 | 37.9 | 38.6 |
| CF (%vol) | 2 | 2 | 0 | 0 |
| CaCO$_3$ tot (dag kg$^{-1}$) | 2.4 | 2.5 | 2.3 | 2.1 |
| SOC (dag kg$^{-1}$) | 31.4 | 11.0 | 9.0 | 7.2 |
| TN (g kg$^{-1}$) | 24.4 | 9.8 | 9.7 | 7.7 |
| AWC (mm) | 12 | 45 | 84 | 70 |
| Cluster 2 | Soil Horizons | | | |
| (Profile 4) | Ap1 | Ap2 | Bt | Btg |
| Depth (m) | 0.10 | 0.35 | 0.90 | 1.15 |
| Clay (dag kg$^{-1}$) | 18.4 | 16.1 | 16.8 | 24.7 |
| Silt (dag kg$^{-1}$) | 37.3 | 35.4 | 36.9 | 35.6 |
| Sand (dag kg$^{-1}$) | 44.3 | 48.5 | 46.3 | 39.7 |
| CF (%vol) | 12 | 12 | 5 | 0 |
| CaCO$_3$ tot (dag kg$^{-1}$) | 2.4 | 2.3 | 2.3 | 2.3 |
| SOC (dag kg$^{-1}$) | 9.0 | 13.1 | 9.0 | 6.8 |
| TN (g kg$^{-1}$) | 9.1 | 13.0 | 8.9 | 7.5 |
| AWC (mm) | 11 | 28 | 66 | 33 |

*3.2. Irrigation and Cluster Effect on Tree Water Status, Yield and Vegetative Growth*

One day after the beginning of the irrigation period (DOY 187) the tree water status was the same across all irrigation regimes and clusters, whereas significant differences in SWP between irrigation treatments were measured at DOY 219 ($-1.49$, $-2.49$ and $-3.65$ MPa in FI, DI and RF trees, respectively) and DOY 259 ($-1.57$, $-2.65$ and $-3.89$ MPa, respectively) (Table 2). Similar trends were measured in both clusters without any significant interaction between irrigation treatment and cluster on tree water status (data not shown).

The fruit yield per tree and the TCSA increment were affected by irrigation, canopy NDVI, and soil ECa$_1$. Irrigation affected both fruit yield (23.890, 15.745 and 8.040 kg per tree in FI, DI, and RF trees, respectively) and the TCSA increment (0.37, 0.23 and 0.16 dm$^2$, respectively) (Figure 4). The two clusters obtained using both remote (NDVI) and proximal (ECa$_1$) sensed indices showed differences in fruit yield (17.259 and 14.003 kg per tree in Cluster 1 and 2, respectively) and annual TCSA increment (0.26 and 0.24 dm$^2$, respectively) (Figure 4). The effect of irrigation on tree productivity and vegetative growth was different between the two clusters. The trees under deficit irrigation grown in the orchard area associated to Cluster 1 showed yields similar to those measured in fully irrigated trees of Cluster 2. Similarly, rainfed trees in Cluster 1 had yields similar to those of trees under deficit irrigation in Cluster 2. The vegetative growth showed significant differences between all the irrigation-cluster combinations (Figure 4).

**Table 2.** Stem water potential measured on olive trees grown in different orchard zones identified as Cluster 1 (C1), and Cluster 2 (C2) and subjected to full irrigation, deficit irrigation and rainfed conditions. Values are means ± standard deviation of six trees per irrigation treatment and three trees per each cluster-irrigation regime combination. Different letters indicate significant differences within each cluster after ANOVA ($p < 0.05$). DOY 187: one day before the beginning of the differentiation of the irrigation treatments.

| Cluster | Irrigation | Stem Water Potential (MPa) | | |
| --- | --- | --- | --- | --- |
| | | DOY 187 | DOY 219 | DOY 259 |
| C1 + C2 | Full | −1.43 ± 0.07 | −1.49 ± 0.06 a | −1.57 ± 0.06 a |
| | Deficit | −1.39 ± 0.09 | −2.49 ± 0.13 b | −2.65 ± 0.07 b |
| | Rainfed | −1.46 ± 0.07 | −3.65 ± 0.09 c | −3.89 ± 0.11 c |
| C1 | Full | −1.43 ± 0.08 | −1.50 ± 0.09 a | −1.58 ± 0.09 a |
| | Deficit | −1.43 ± 0.10 | −2.55 ± 0.05 b | −2.64 ± 0.11 b |
| | Rainfed | −1.42 ± 0.08 | −3.60 ± 0.05 c | -3.88 ± 0.08 c |
| C2 | Full | −1.42 ± 0.08 | −1.48 ± 0.03 a | −1.56 ± 0.03 a |
| | Deficit | −1.38 ± 0.12 | −2.45 ± 0.13 b | −2.63 ± 0.00 b |
| | Rainfed | −1.50 ± 0.05 | −3.70 ± 0.10 c | −3.90 ± 0.15 c |

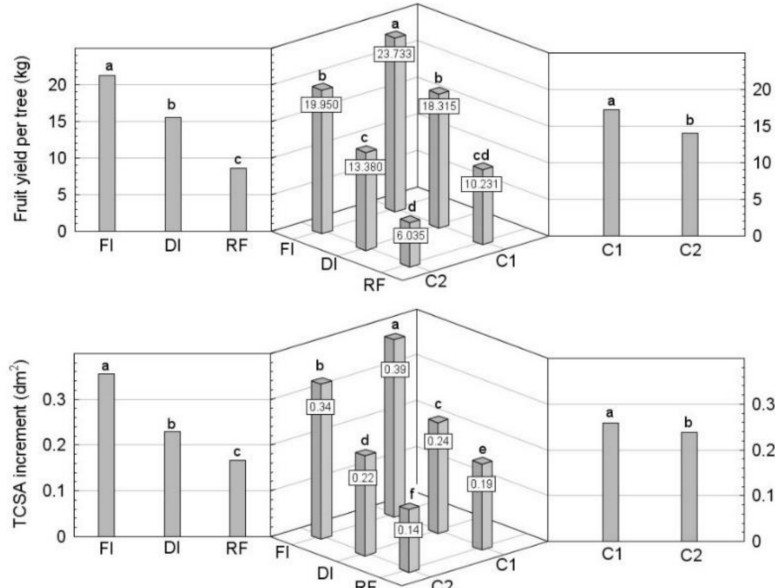

**Figure 4.** Fruit yield and trunk cross sectional area (TCSA) increment in olive trees subjected to different irrigation regimes (Full irrigation, FI; deficit irrigation, DI; rainfed conditions, RF) and located in different orchard zones (Cluster 1, C1; Cluster 2, C2). Histograms represent the average of 18 (FI), 40 (DI), 15 (RF), 33 (C1), 39 (C2), 7 (FI-C1), 11 (FI-C2), 17 (DI-C1), 23 (DI-C2), 9 (RF-C1), 6 (RF-C2) trees per each irrigation treatment and cluster combination. Different letters indicate significant differences of LSD test after ANOVA ($p < 0.05$).

The highest values of $WUE_f$ and $WUE_o$ were measured in DI trees in Cluster 1, whereas in Cluster 2 the WUE (both fruit and oil) increased with the amount of water used by the tree (Table 3). Differences in WUE between clusters emerged within each irrigation regime and became increasingly evident as the level of water deficit increased (Table 3).

**Table 3.** Water use efficiency of fruit ($WUE_f$) and oil ($WUE_o$) production measured in olive trees grown in different orchard zones (Cluster 1, C1, and Cluster 2, C2) and subjected to full irrigation (FI), deficit irrigation (DI) and rainfed conditions (RF). $WUE_f$ and $WUE_o$ are the ratio between fruit (dry weight) or oil yield, and annual crop evapotranspiration ($ET_c$).

| Irrigation | $WUE_f$ (g Dry Weight $L^{-1}$ $H_2O$) | | | $WUE_o$ (g Oil $L^{-1}$ $H_2O$) | | |
|---|---|---|---|---|---|---|
| | C1 + C2 | C1 | C2 | C1 + C2 | C1 | C2 |
| FI | 0.91 | 0.95 | 0.88 | 0.36 | 0.38 | 0.35 |
| DI | 0.97 | 1.15 | 0.78 | 0.42 | 0.49 | 0.34 |
| RF | 0.66 | 0.82 | 0.50 | 0.27 | 0.34 | 0.19 |

*3.3. Comparing Proximal and Remote Sensing Indices against Tree Performances*

Significant correlations were evident between NDVI and fruit yield (r = 0.54) and between NDVI and the annual TCSA increment (r = 0.33) when the trees from all irrigation treatments were considered together (Figure 5). Among the soil indices derived from soil sensors, only $ECa_2$ showed a significant, albeit lower, correlation with fruit yield (r = 0.29). Within each irrigation treatment, the highest yield was measured in the most vigorous trees (highest canopy NDVI values) grown in areas with the highest soil $ECa_1$ values, even though a different impact of NDVI and soil $ECa_1$ was observed according to the irrigation treatment (Figure 6).

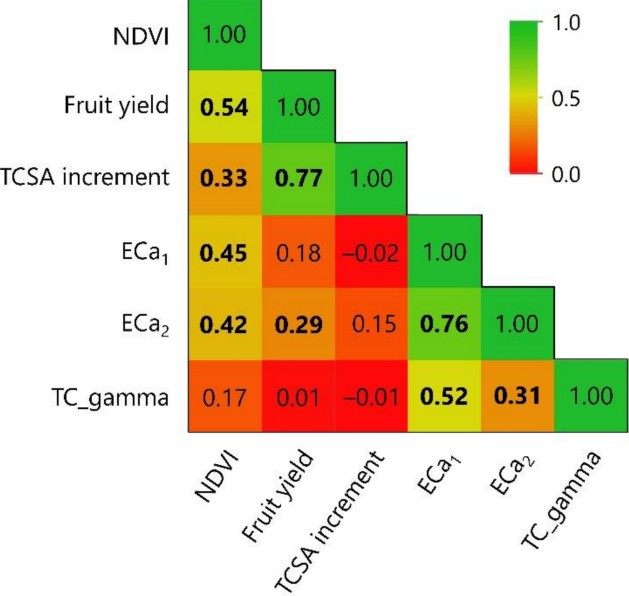

**Figure 5.** Nonparametric correlation coefficients (Spearman's ranks) between NDVI, tree parameters (fruit yield and annual TCSA increment) and soil indices ($ECa_1$, $ECa_2$ and TC_gamma) derived by proximal sensing sensors at the experimental site. In bold, the coefficient significant for $p < 0.01$ (n = 75, all irrigation treatments).

In particular, the relationship between NDVI and fruit yield and between NDVI and the annual TCSA increment in fully-irrigated trees produced r values of 0.71 and 0.77, respectively, whereas $ECa_1$ showed lower, but still significant correlations (r = 0.62 for both fruit yield and annual TCSA increment) (data not shown). Different results were obtained under rainfed conditions, where the ECa1 showed a higher level of correlation (r = 0.67) with fruit yield than that obtained for the relationship between NDVI and fruit yield (r = 0.54) (data not shown).

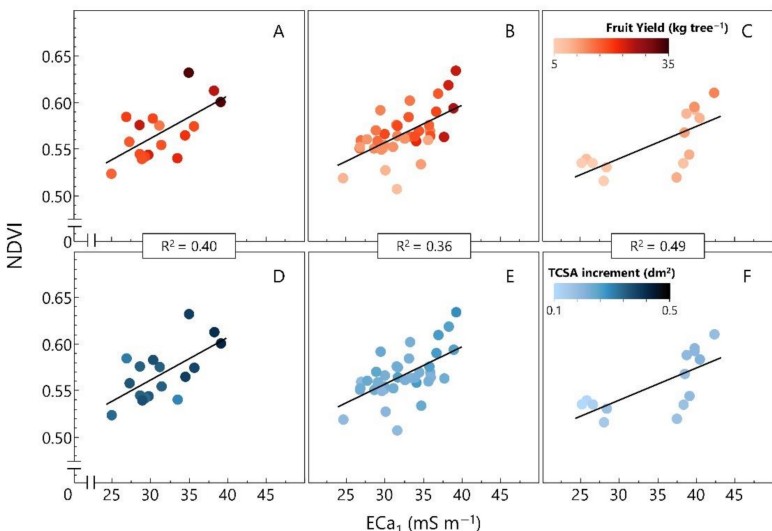

**Figure 6.** The relationship between soil $ECa_1$ and the NDVI of trees grown in the soil area under full irrigation (**A,D**), deficit irrigation (**B,E**), and rainfed conditions (**C,F**). Each point represents one tree (and the respective allocated soil area). Coefficients of determination were calculated within each irrigation treatment. Regression equations: $NDVI = 0.42 + 0.005\ ECa_1$ (**A,D**); $NDVI = 0.44 + 0.004\ ECa_1$ (**B,F**); $NDVI = 0.44 + 0.003\ ECa_1$ (**C,F**).

A stepwise regression analysis was used to highlight the impact of NDVI, $ECa_1$, $ECa_2$, and TC_gamma on fruit yield and TCSA increment prediction within each irrigation regime (Table 4; Table 5). The regression results show that, under full irrigation, the only variable predictive of yield and TCSA increment was NDVI. Under deficit irrigation, soil spatial variability, determined by ECa maps, also played an important role on yield and TCSA increment, although NDVI remained the most important variable for the prediction of these olive tree parameters. Under rainfed conditions, NDVI was not useful to predict yield, whereas soil features, defined by $ECa_1$ and TC_gamma, on yield, became significant explanatory variables in the multiple regression. On the other hand, for the prediction of TCSA increment, NDVI remained a significant explanatory variable, together with $ECa_1$ and $ECa_2$ under rainfed conditions.

**Table 4.** Results of forward stepwise regressions calculated for the fruit yield and the three irrigation regimes (FI: Full irrigated; DI: Deficit irrigation; RF: Rainfed, no irrigation). Legend: B, coefficient of the regression; *p*-value, value of the *t*-test of the coefficient, $R^2_{adj}$, adjusted $R^2$; F, F-test value of the multiple regression and degrees of freedom; SEP, Standard Error of Prediction.

| Irrigation | Predictive Variables | B | *p*-Value (*t*-Test) | $R^2_{adj}$ | F (df) | *p*-Value (Regression) | SEP (kg/Tree) |
|---|---|---|---|---|---|---|---|
| FI | Intercept<br>NDVI | −36.7<br>103.2 | 0.021<br><0.001 | 0.48 | 16.5 (16) | 0.001 | 3.92 |
| DI | Intercept<br>NDVI<br>$ECa_1$ | −48.1<br>85.5<br>0.5 | <0.001<br><0.001<br>0.002 | 0.61 | 31.9 (37) | <0.001 | 2.96 |
| RF | Intercept<br>$ECa_1$<br>TC_gamma | 100.5<br>0.54<br>−0.26 | 0.038<br>0.001<br>0.035 | 0.55 | 10.7 (14) | 0.001 | 1.98 |

**Table 5.** Results of forward stepwise regressions calculated for the trunk cross sectional area increment and the three irrigation regimes (FI: Full irrigated; DI: Deficit irrigation; RF: Rainfed, no irrigation). Legend: B, coefficient of the regression; *p*-value, value of the *t*-test of the coefficient; $R^2_{adj}$, adjusted $R^2$; F, F-test value of the multiple regression and degrees of freedom; SEP, Standard Error of Pre-diction.

| Irrigation | Predictive Variables | B | *p*-Value (*t*-Test) | $R^2_{adj}$ | F (df) | *p*-Value (Regression) | SEP (dm$^2$) |
|---|---|---|---|---|---|---|---|
| FI | Intercept<br>NDVI | −0.12<br>0.85 | 0.250<br><0.001 | 0.57 | 23.8 (16) | <0.001 | 0.027 |
| DI | Intercept<br>NDVI<br>ECa$_2$ | −0.15<br>0.51<br>0.01 | 0.017<br><0.001<br>0.059 | 0.49 | 20.1 (37) | <0.001 | 0.018 |
| RF | Intercept<br>NDVI<br>ECa$_1$<br>ECa$_2$ | 0.07<br>0.43<br>0.01<br>−0.01 | 0.636<br>0.033<br>0.016<br>0.072 | 0.61 | 9.5 (13) | 0.001 | 0.02 |

## 4. Discussion

### 4.1. Irrigation and Cluster Effect on Tree Performance

Increasing volumes of irrigation increased yield and TCSA of olive trees, in agreement with the large literature on the subject [5,7–9,48]. Differences between irrigation treatments were significant regardless of the different soil types in the orchard, as expected. Previous studies on field crops also reported that, although soil properties were major sources of plant variability, water management and fertilization had greater impact on vegetative activity than soil features [34,65]. Precision management of olive orchards aims at optimizing the water use efficiency in terms of fruit (WUE$_f$) and oil (WUE$_o$) yield per liter of consumed water. We determined the impact of the three irrigation regimes on tree productivity in the two clusters identified by the combined use of soil proximal and vegetative remote sensing. The values of WUE$_o$ measured in our work were much higher than those measured by Fernandes-Silva et al. [9] in a similar experiment carried out in Portugal (WUE$_o$ of 0.19, 0.16 and 0.07 g L$^{-1}$ for T2-100% ETc, T1-30% ETc and T0-rainfed, respectively), but similar to those reported by Iniesta et al. [8] (WUE$_o$ of 0.24 and 0.30 for full and sustained deficit irrigation, respectively). We showed differences in either WUE$_f$ or WUE$_o$ within the orchard. The higher tree productivity measured in Cluster 1 also resulted in a higher WUE$_f$ (0.90) and WUE$_o$ (0.32) compared to Cluster 2 (0.67 and 0.27 for WUE$_f$ and WUE$_o$, respectively). In particular, in Cluster 1 the WUE$_f$ and WUE$_o$ of FI trees was 0.98 and 0.39, respectively, whereas in Cluster 2 the trees subjected to the same irrigation regime showed values of 0.90 and 0.36 for the same parameters, respectively. Differences in WUE$_f$ and WUE$_o$ in the two clusters were more evident under deficit irrigation (WUE$_f$ and WUE$_o$ of 1.06 and 0.38, respectively, in Cluster 1, and 0.72 and 0.31 in Cluster 2), and rainfed conditions (WUE$_f$ and WUE$_o$ of 0.66 and 0.21 g L$^{-1}$, respectively, in Cluster 1, and 0.40 and 0.15 g L$^{-1}$ in Cluster 2), indicating the increasing impact of the cluster characteristics on tree productivity as irrigation volumes decreased. In a recent study carried out in a high density olive orchard the two management zones identified using a similar approach (combining proximal and remote sensing) showed significant differences in fruit yield only in one out of three years [46]. The authors reported that the lack of differences in those two years was due to a different irrigation management (timing and amount of the applied water) in the two zones.

### 4.2. Comparing Proximal and Remote Sensing Indices against Tree Performance

The ability of proximal and remote sensing in estimating tree growth and yield has been poorly investigated in olive growing so far. Few previous studies used UAV imagery to estimate fruit yield and vegetative growth of olive trees [25,26,32,33]. In the current work, we observed a general (all irrigation treatments) linear relationship between NDVI and fruit

yield (r = 0.54), similarly to previous findings in a hedgerow olive orchard [26]. In that same study, there was no relationship between NDVI and fruit yield in the year when abiotic (spring frost) and biotic (olive fruit fly) factors dramatically affected final fruit yield [26]. A specific response to location and growing season in the NDVI-fruit yield relationship also appeared when results obtained in similar experiments were compared [32,33], highlighting the limit of the fruit yield prediction carried out several months before harvest. A lower level of correlation (r = 0.33) was observed between NDVI and the TCSA increment when all the trees from the three irrigation treatments were considered together. In a previous study carried out using the same cultivar, Caruso et al. [25] reported a combined effect of tree water status and canopy NDVI on the final TCSA increment. Soil indices derived by proximal sensing ($ECa_1$, $ECa_2$, and TC_gamma) were less able to predict fruit yield (r = 0.18, 0.29 and 0.01, respectively) and TCSA increment (0.02, 0.15 and 0.01) when all irrigation treatments were considered together. A wide range of correlation coefficients between ECa and fruit yield ($R^2$ comprised between 0.01 and 0.94) was reported in previous studies carried out on different cultivars of apple trees [44,66]. In a previous study carried out in a commercial olive orchard, significant relationships between the soil characteristics (organic matter, B and Ca) derived by a systematic sampling grid and the fruit yield were measured only in one of the two experimental years [67].

To discriminate the impact of proximally (soil) and remotely (tree) determined indices on tree parameters (yield and TCSA increment) under different irrigation regimes we used a stepwise multiple regression analysis. Under full irrigation conditions, the NDVI was the only variable able to describe the variability in fruit yield and vegetative growth, whereas under rainfed conditions the soil parameters ($ECa_1$, $ECa_2$ and TC_gamma) were identified as the only factors affecting yield. Interestingly, under deficit irrigation, both tree (NDVI) and soil ($ECa_1$ and $ECa_2$) parameters were identified as factors contributing to the measured yield parameters. In a similar study in an irrigated vineyard, Terrón et al. [68] also reported high correlations between NDVI and soil ECa, but such a relationship was strongly dependent on the growing season and the irrigation regime. Therefore, the role of soil spatial variability was much more important as water deficit became more severe, whereas in cases when water availability was high due to rains or irrigation, the variability of soil characteristics played a less important role on vegetative growth. Priori et al. [69] showed that the effects of soil features on grape and wine peculiarities were much more evident during dry years, whereas rainy summers tended to hide this variability in a rainfed vineyard.

## 5. Conclusions

We showed that remote and proximal sensing technologies allowed to determine that the effect of different irrigation regimes on tree performance and WUE depended on the location within the orchard (Cluster 1 and Cluster 2). The second finding of this study concerns the major role of tree vigour in determining the final fruit yield under optimal soil water availability, whereas soil features become the most important factors under rainfed conditions. Since water availability in the Mediterranean climate is becoming increasingly limited, this information may be crucial to optimize the water management through site-specific irrigation protocols. They should be considered preliminary, as the investigation was done over one growing season only. Furthermore, since olive trees are subject to alternate bearing, long-term studies are needed to evaluate the stability of the orchard zone effects on yield and tree growth in high- ("on-year") and low- ("off-year") yield crop years.

**Author Contributions:** G.C.: conceptualization, investigation, methodology, formal analysis, resources, software, writing—original draft and editing. G.P.: investigation, data curation, formal analysis, software, writing—review and editing. R.G.: resources, supervision, validation and writing—review. S.P.: conceptualization, investigation, methodology, formal analysis, resources, software and writing—original draft. All authors have read and agreed to the published version of the manuscript.

**Funding:** This research received no external funding.

**Institutional Review Board Statement:** Not applicable.

**Informed Consent Statement:** Not applicable.

**Data Availability Statement:** Not applicable.

**Acknowledgments:** The irrigation system was kindly supplied by Netafim Italia.

**Conflicts of Interest:** The authors declare no conflict of interest.

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
