# Peer review of "Remote and Proximal Sensing Techniques for Site-Specific Irrigation Management in the Olive Orchard"

_applsci, doi:10.3390/app12031309_

Round 1

Reviewer 1 Report

Based on the application of remote and proximal sensing techniques in an olive orchard, this manuscript describes the experimental realization of site specific irrigation management in an olive orchard. Despite the fact that this is a highly significant topic, I think the paper has some flaws, particularly in terms of some data analysis and writing, and I feel the analysis has not been carried out to its maximum potential. The experimental location selection is one of my primary concerns. Considering that the authors have focused on a relatively small area, it will be difficult to present scientific proof on the basis of such a narrow interpretation. The authors have employed two separate proximal soil sensors (PSS) to map soil spatial variability without validating or addressing the validity of these two sensors for this specific site. I recommend that the paper be rejected due to its flaws. My detailed comments are listed below:

  1. The introduction portion does not provide enough background information.
  2. The study's motivation was unclear. There was no connection between the paragraphs.
  3. The authors may choose to include examples of some of the applications of precision irrigation technologies in order to make their motive apparent and to separate their research from other papers.
  4. For this detailed experimental investigation, authors should justify their selection of a small scale experimental location.
  5. The influence of both proximal sensors' height/distance on soil surface and subsurface properties was not addressed in the text.
  6. The authors conducted interpolation for observation data using Ordinary Kriging. Why this interpolation method was selected for this study? A detail explanation of the interpolation need to be explained clearly in the text.
  7. Green and blue colors are used to indicate Cluster 1 and Cluster 2, respectively; however, no explanation was given for the pink color cluster.
  8. What criteria did you use to create the clusters for this experiment?
  9. Why was soil zoning done if the plots/treatments were not delineated/conducted considering the soil zones?
  10. Line 120: What is RF? At first, please try to write abbreviate.
  11. How 513 trees were divided in 12 zones? Not clearly describe in the text.
  12. The total area was divided into three different types of irrigation methods, why the authors have chosen different number of plots for each type of irrigation technique? They have chosen only two plots for RF, three plots for FI, and the remaining plots were considered for DI. Please provide justification for your selection procedure.
  13. What were the criteria that were used for the deployment of the six sensors? As presented in Figure 2.
  14. Results of only RF treatment are discussed in the text, please also elaborate the results of DI and FI.
  15. Please provide a separate conclusion section.

Overall, I am not sure that the results will be judged novel or important enough for publication in Applied Sciences.

Author Response

Reviewer #1

Based on the application of remote and proximal sensing techniques in an olive orchard, this manuscript describes the experimental realization of site specific irrigation management in an olive orchard. Despite the fact that this is a highly significant topic, I think the paper has some flaws, particularly in terms of some data analysis and writing, and I feel the analysis has not been carried out to its maximum potential. The experimental location selection is one of my primary concerns. Considering that the authors have focused on a relatively small area, it will be difficult to present scientific proof on the basis of such a narrow interpretation. The authors have employed two separate proximal soil sensors (PSS) to map soil spatial variability without validating or addressing the validity of these two sensors for this specific site. I recommend that the paper be rejected due to its flaws. My detailed comments are listed below:

Authors: The reviewer # 1 reported major flaws in the data analysis and interpretation, and writing of the manuscript. His/her constructive suggestions were however used to improve the manuscript, despite his/her rejection. One of the main concerns of reviewer #1 is the experimental location and limited size of the area. The reason for using a small orchard is that all trees were monitored individually since the year of planting (2003). This means that vegetating growth, pruning material, yield, product quality, water and fertilizers supplied were monitored for every single tree since planting. In addition, every five years root growth was monitored by soil coring or soil trenching. Climatic variables were also recorded on site for over 15 years. This level of detail made us confident about the interpretation of results obtained out of the present work. We believe that this information (representing the ground truth data) is important in experiments aimed at testing new technologies and that the results we obtained provide new evidence about proximal and soil sensing in olive growing. However, we are aware that the study area is small, and that results cannot be generalized and should be considered preliminary.

Reviewer #1: The introduction portion does not provide enough background information. The study's motivation was unclear. There was no connection between the paragraphs.

Authors: The paper is already quite long. We repute the background information sufficient for the scope of the paper. The list of references is already very long.

Reviewer #1: The authors may choose to include examples of some of the applications of precision irrigation technologies in order to make their motive apparent and to separate their research from other papers.

Authors: We have improved this section according to the reviewer’s suggestion (lines 76-80)

Reviewer #1: For this detailed experimental investigation, authors should justify their selection of a small scale experimental location.

Authors: see general reply above

Reviewer #1: The influence of both proximal sensors' height/distance on soil surface and subsurface properties was not addressed in the text.

Authors: We added in the text “with the sensors at a height from the soil surface around 20 cm, to minimize the signal attenuation”.

Reviewer #1: The authors conducted interpolation for observation data using Ordinary Kriging. Why this interpolation method was selected for this study? A detail explanation of the interpolation need to be explained clearly in the text.

Authors: Ordinary kriging is commonly used to interpolate soil proximal sensing data, because such types of data is perfect for this type of interpolation. With soil proximal sensing data, the spatial frequency of the data is very high (thousands of points per hectare), therefore the selection of different interpolation methods does not bring relevant improvements. However, we added in the text: “which is widely used or the interpolation of soil proximal sensing data (some references)”

Reviewer #1: Green and blue colors are used to indicate Cluster 1 and Cluster 2, respectively; however, no explanation was given for the pink color cluster.

Authors: We have improved Fig. 1 and its description. The pink coloured trees were not considered in our analysis as they belonged to cultivars other than Frantoio

Reviewer #1: What criteria did you use to create the clusters for this experiment?

Authors: The criteria and the methods have been described in the chapter 2.5 “Experimental design and statistical analysis”:“The orchard was subdivided into two homogeneous zones (C1 and C2) by k-means clustering, using the Hill-Climbing algorithm [63] of the SAGA-GIS software. The input data to calculate k-means clustering were: ECa1, ECa2, TC_gamma, and NDVI.” We added in the text the reason to select only two clusters: “The selection of only two clusters was driven by the small surface of the experimental field and by the scarce heterogeneity of the soil.”

Reviewer #1: Why was soil zoning done if the plots/treatments were not delineated/conducted considering the soil zones?

Authors: The soil zones were delineated after the establishing of the irrigation system and the four independent sectors.

Reviewer #1: Line 120: What is RF? At first, please try to write abbreviate.

Authors: Rainfed. Done.

Reviewer #1: How 513 trees were divided in 12 zones? Not clearly describe in the text.

Authors: The orchard consisted in 140 trees. 513 is the number of trees per hectare (tree density) calculated using the tree spacing (5m x 3.9m). Thus, 10.000 m2 / 19.5 m2 = 512.8 (approximated to 513) trees per hectare.

Reviewer #1: The total area was divided into three different types of irrigation methods, why the authors have chosen different number of plots for each type of irrigation technique? They have chosen only two plots for RF, three plots for FI, and the remaining plots were considered for DI. Please provide justification for your selection procedure.

Authors: The orchard is constituted by 12 plots. Each plot is constituted by three rows (with the only exception of one RF plot, constituted by two rows). For simplicity, adjacent plots of the same irrigation treatment were not separated in Fig. 1. The irrigation system is constituted by four sector. Each sector provide water to three plots. We decided to use two sectors for the deficit irrigation (DI) treatment because it is the most promising irrigation strategy in terms of fruit yield and oil quality.

Reviewer #1: What were the criteria that were used for the deployment of the six sensors? As presented in Figure 2.

Authors: We do not understand. What do you mean for “sensors”?

Reviewer #1: Results of only RF treatment are discussed in the text, please also elaborate the results of DI and FI.

Authors: Results of DI and FI treatments are already reported in the text (Lines: 314-315; 319-321; 325-329; 429-437).

Reviewer #1: Please provide a separate conclusion section.

Authors: Done

Reviewer #1: Overall, I am not sure that the results will be judged novel or important enough for publication in Applied Sciences.

Reviewer 2 Report

Dear Authors,

In my opinion, the article „Remote and proximal sensing techniques for site-specific irrigation management in the olive orchard” meets all the criteria to be published in this journal. However, it does require minor revision:

L10-23: The abstract should be a single paragraph without headings: Background, Methods, Results, Conclusion.

L101-104: Why were trees grouped into 2 clusters and on what criteria was the clustering performed? Also, why are other trees that are not included in any cluster? I saw a description of the clusters on lines 207-209, but it's too short. I would also suggest moving the description before Figure 1 to better understand what these elements represent on the map. Moreover, I would suggest that in figure 1 you add an image with the location of these plots at regional level and/or geographical coordinates.

The paragraph between lines 198 and 221, presents several statistical methods, please consider the division into smaller paragraphs for each method. It is difficult to read in this form.

L203: What is the difference between ECa1 and ECa2? What does TC_gamma mean? I found the explanation in Chapter 3 Results, but I would suggest that it be moved to where the acronym first appears in the text. Same for TC_gamma.

L252: Table 1 should be inserted after quoting on lines 255-257. Also, the explanation for Ap1 and Ap2 is missing from the legend. And I would suggest the legend be added below the table (table footer).

L270: Also, Table 2 should be inserted after quoting and the legend should be added below the table (table footer).

L339-342: The paragraph should be moved before Table 3.

L344-348: Figure 4 should be moved after citation.

L394-460: The paragraphs are very long. I would suggest that they be divided into smaller ones to make the text easier to follow.

L461-472: The conclusions should be set out in a separate chapter.

All the best!

Author Response

Reviewer #2

Dear Authors,

In my opinion, the article “Remote and proximal sensing techniques for site-specific irrigation management in the olive orchard” meets all the criteria to be published in this journal. However, it does require minor revision:

Reviewer #2: L10-23: The abstract should be a single paragraph without headings: Background, Methods, Results, Conclusion.

Authors: Thank you for your suggestion. Done.

Reviewer #2: L101-104: Why were trees grouped into 2 clusters and on what criteria was the clustering performed?

Authors: The criteria and the methods are described in paragraph 2.5 “Experimental design”. “The orchard was subdivided into two homogeneous zones (C1 and C2) by k-means clustering, using the Hill-Climbing algorithm [63] of the SAGA-GIS software. The input data to calculate k-means clustering were: ECa1, ECa2, TC_gamma, and NDVI.” However, we added in the text the reason to select only two clusters: “The selection of only two clusters was driven by the small surface of the experimental field and by the relatively homogeneous soil.”

Reviewer #2: Also, why are other trees that are not included in any cluster?

Authors: The trees not included in the experiment belonged to cultivars other than Frantoio.

Reviewer #2: I saw a description of the clusters on lines 207-209, but it's too short. I would also suggest moving the description before Figure 1 to better understand what these elements represent on the map.

Authors: We moved the Fig.1 at the end of the paragraph, and we have explained the reasons of the selection of two clusters.

Reviewer #2: Moreover, I would suggest that in figure 1 you add an image with the location of these plots at regional level and/or geographical coordinates.

Authors: Thank you for your suggestion. Done.

Reviewer #2: The paragraph between lines 198 and 221, presents several statistical methods, please consider the division into smaller paragraphs for each method. It is difficult to read in this form.

Authors: Thank you for your suggestion. We have subdivided the paragraph into: “Experimental design” and “Statistical analysis”.

Reviewer #2: L203: What is the difference between ECa1 and ECa2? What does TC_gamma mean? I found the explanation in Chapter 3 Results, but I would suggest that it be moved to where the acronym first appears in the text. Same for TC_gamma.

Authors: Thank you for your suggestion. We reported the explanation where the acronym first appears. In addition, we better explain the meaning of TC gamma and the method for the elaboration of gamma spectroscopy.

Reviewer #2: L252: Table 1 should be inserted after quoting on lines 255-257. Also, the explanation for Ap1 and Ap2 is missing from the legend. And I would suggest the legend be added below the table (table footer).

Authors: Thank you for your suggestion. We have added the description of Ap1 and Ap2. In previous papers published in Applied Sciences the legend is located above the table and, therefore, unless different indication by the Editor, we prefer to keep it in the current position.

Reviewer #2: L270: Also, Table 2 should be inserted after quoting and the legend should be added below the table (table footer).

Authors: Done. For the legend see the previous reply.

Reviewer #2: L339-342: The paragraph should be moved before Table 3.

Authors: Done.

Reviewer #2: L344-348: Figure 4 should be moved after citation.

Authors: Done.

Reviewer #2: L394-460: The paragraphs are very long. I would suggest that they be divided into smaller ones to make the text easier to follow.

Authors: Thank you for your suggestion. We separated this paragraph into two subsections.

Reviewer #2: L461-472: The conclusions should be set out in a separate chapter.

Authors: Thank you, done.

All the best!

Reviewer 3 Report

My recommendation about the manuscript is accept after minor revision.

1) In the manuscript (Remote and Proximal Sensing Applied to Agriculture and Forest Sciences) the authors should provide a flow chart for the study. I think flow chart will help to improve the methodology section. 

2) The authors should provide a study area figure and add study area photographs from field work. 

3) The authors should provide a clear explanation about differences of the study than the published ones. 

Author Response

Reviewer #3

My recommendation about the manuscript is accept after minor revision.

Reviewer #3: In the manuscript (Remote and Proximal Sensing Applied to Agriculture and Forest Sciences) the authors should provide a flow chart for the study. I think flow chart will help to improve the methodology section. 

Authors: Thank you for your suggestion. We have added a flow chart in M&M.

Reviewer #3: The authors should provide a study area figure and add study area photographs from field work. 

Authors: We have added a photograph from field work as suggested.

Reviewer #3: The authors should provide a clear explanation about differences of the study than the published ones. 

Authors: We improved the introduction paragraph to better highlight differences with other studies on the same subject.
